# Polyvalent Bacterial Lysate with Potential Use to Treatment and Control of Recurrent Urinary Tract Infections

**DOI:** 10.3390/ijms25116157

**Published:** 2024-06-03

**Authors:** Salvador Eduardo Acevedo-Monroy, Luz María Rocha-Ramírez, Daniel Martínez Gómez, Francisco Javier Basurto-Alcántara, Óscar Medina-Contreras, Ulises Hernández-Chiñas, María Alejandra Quiñones-Peña, Daniela Itzel García-Sosa, José Ramírez-Lezama, José Alejandro Rodríguez-García, Edgar González-Villalobos, Raúl Castro-Luna, Leonel Martínez-Cristóbal, Carlos Alberto Eslava-Campos

**Affiliations:** 1Laboratorio de Patogenicidad Bacteriana, Unidad de Hemato-Oncología e Investigación, Hospital Infantil de México Federico Gómez/Facultad de Medicina, Universidad Nacional Autónoma de México, Dr. Márquez No. 162, Col Doctores, Alcaldía Cuauhtémoc, Ciudad de México 06720, Mexico or salcevedom@gmail.com (S.E.A.-M.); ulisesh@unam.mx (U.H.-C.); maria.quinones@utrgv.edu (M.A.Q.-P.); danielaitzelgs@hotmail.com (D.I.G.-S.); alej.rodriguez42@gmail.com (J.A.R.-G.); 2Laboratorio de Microbiología Molecular Departamento de Microbiología e Inmunología, Facultad de Medicina Veterinaria y Zootecnia, Universidad Nacional Autónoma de México, Av. Universidad #3000, Colonia, C.U., Coyoacán, Ciudad de México 04510, Mexico; 3Unidad de Investigación en Enfermedades Infecciosas, Hospital Infantil de México Federico Gómez. Secretaría de Salud, Dr. Márquez No. 162, Col Doctores, Alcaldía Cuauhtémoc, Ciudad de México 06720, Mexico; luzmrr7@yahoo.com.mx; 4Departamento de Producción Agrícola y Animal, Laboratorio de Microbiología Agropecuaria, Universidad Autónoma Metropolitana Xochimilco, Calzada del Hueso 1100, Colonia Villa Quietud, Alcaldía Coyoacán, C.P., Ciudad de México 04960, Mexico; dmartinez@correo.xoc.uam.mx; 5Laboratorio de Vacunología y Constatación, Departamento de Microbiología e Inmunología, Facultad de Medicina Veterinaria y Zootecnia, Universidad Nacional Autónoma de México, Av. Universidad #3000, Colonia, C.U., Coyoacán, Ciudad de México 04510, Mexico; basurto@unam.mx; 6Unidad de Investigación Epidemiológica en Endocrinología y Nutrición, Hospital Infantil de México Federico Gómez, Dr. Márquez No. 162, Col. Doctores, Alcaldía Cuauhtémoc, Ciudad de México 06720, Mexico; omedina@himfg.edu.mx; 7Unidad Periférica de Investigación Básica y Clínica en Enfermedades Infecciosas; Departamento de Salud Pública, División de Investigación Facultad de Medicina, Universidad Nacional Autónoma de México, Dr. Márquez No. 162, Col Doctores, Alcaldía Cuauhtémoc, Ciudad de México 06720, Mexico; 8Department of Health & Biomedical Science College of Health Professions, Biomedical Science, The University of Texas Rio Grande Valley, Edinburg, TX 78539, USA; 9Departamento de Patología, Facultad de Medicina Veterinaria y Zootecnia, Universidad Nacional Autónoma de México, Ciudad de México 04510, Mexico; pepeton_60@yahoo.com.mx; 10Laboratorio de Epidemiología Molecular, Departamento de Salud Pública División de Investigación Facultad de Medicina, Universidad Nacional Autónoma de México, Av. Universidad #3000, Colonia, C.U., Coyoacán, Ciudad de México 04510, Mexico; edgar.villalobos@comunidad.unam.mx; 11Bioterio, Hospital Infantil de México Federico Gómez, Dr. Márquez No. 162, Col Doctores, Alcaldía Cuauhtémoc, Ciudad de México 06720, Mexico; raulcaslu@yahoo.com.mx (R.C.-L.); martinezcristobal@yahoo.com.mx (L.M.-C.)

**Keywords:** urinary tract infections, bacterial lysates, animal model, in vitro assays, immunostimulant, culture cells

## Abstract

Overuse of antimicrobials has greatly contributed to the increase in the emergence of multidrug-resistant bacteria, a situation that hinders the control and treatment of infectious diseases. This is the case with urinary tract infections (UTIs), which represent a substantial percentage of worldwide public health problems, thus the need to look for alternatives for their control and treatment. Previous studies have shown the usefulness of autologous bacterial lysates as an alternative for the treatment and control of UTIs. However, a limitation is the high cost of producing individual immunogens. At the same time, an important aspect of vaccines is their immunogenic amplitude, which is the reason why they must be constituted of diverse antigenic components. In the case of UTIs, the etiology of the disease is associated with different bacteria, and even *Escherichia coli*, the main causal agent of the disease, is made up of several antigenic variants. In this work, we present results on the study of a bacterial lysate composed of 10 serotypes of *Escherichia coli* and by *Klebsiella pneumoniae*, *Klebsiella aerogenes*, *Enterococcus faecalis*, *Proteus mirabilis*, *Citrobacter freundii*, and *Staphylococcus haemolyticus*. The safety of the compound was tested on cells in culture and in an animal model, and its immunogenic capacity by analysing in vitro human and murine macrophages (cell line J774 A1). The results show that the polyvalent lysate did not cause damage to the cells in culture or alterations in the animal model used. The immunostimulatory activity assay showed that it activates the secretion of TNF-α and IL-6 in human macrophages and TNF-α in murine cells. The obtained results suggest that the polyvalent lysate evaluated can be an alternative for the treatment and control of chronic urinary tract infections, which will reduce the use of antimicrobials.

## 1. Introduction

Recently, the increase in antimicrobial multiresistant bacteria has come to pose a serious problem for the treatment and control of infectious diseases. It is estimated that before the middle of the 21st century, millions of people will die because of infectious diseases, or their complications caused by multiresistant bacteria [1]. In this situation, there is a need to look for alternatives that can contribute to the treatment and control of some infectious diseases which, because of the exaggerated and misdirected use of antimicrobials, are becoming more and more difficult to treat. Among these, urinary tract infections (UTIs) have acquired great relevance, both in the number of cases that have increased worldwide and with the different organic and social complications that are associated with this disease [2]. The treatment of UTIs, like that of other infectious diseases, is carried out with antimicrobials; however, the resistance of bacteria to most of the products routinely used for their treatment has limited the options for their control [3,4,5]. Uropathogenic *Escherichia coli* (UPEC) is the most frequently associated bacteria with the pathogenesis of UTIs. However, other bacteria including *Klebsiella pneumoniae*, *Enterococcus faecalis*, *Proteus mirabilis*, and *Staphylococcus* spp, are microorganisms also associated with UTIs, showing high resistance to commonly used antimicrobials for the treatment of this condition [6,7,8,9,10]. Failures in antimicrobial treatment are not the only problem related to the management of UTIs; the use of antimicrobials for long periods can cause alterations in the microbiota and damage to host organs with consequent repercussions such as diabetes, immune response alterations, or inflammatory colitis, among other illnesses [11,12,13,14,15,16,17]. In this situation, it is necessary to implement effective alternative therapies to reduce the use of antimicrobials. During two prospective studies of children and adults with chronic urinary tract infection (UTI), lysates of bacteria isolated from the same patient (autovaccines) were used as an alternative treatment; these were inactivated by heat treatment and sterilized by filtration [18,19]. Both groups of patients were followed for seven months to a year, with urine samples collected every month.

The results of both studies showed a favorable response, controlling the infection in almost 70% of both groups of patients, with in some cases, reinfections occurring between six months and one year after administering the bacterial lysate.

The use of complete microorganisms related to the etiopathogenesis of a condition such as UTI has the advantage of providing diverse immunogenic compounds that can induce an effective polyvalent response [20]. The purpose of the present study was to evaluate the immunostimulatory activity and safety of a polyvalent lysate prepared with previously isolated bacteria [18,19], to have a product with different antigenic components that allows the treatment and control of recurrent urinary tract infections, that is reliable and accessible to the population that requires it and can be used as an alternative to reduce the use of antimicrobials in cases of UTI.

## 2. Results

### 2.1. Selected Bacteria to Elaborate UNAM-HIMFG Lysate

The UNAM-HIMFG polyvalent lysate was elaborated with 10 strains of *E. coli*, 7 of classical serogroups (O25, O75, O6, O8, O1, O16, and O7), and 3 considered as endemic (O9, O17, and O20), and with strains of *Klebsiella pneumoniae*, *Klebsiella aerogenes*, *Citrobacter freundii*, *Proteus mirabilis*, *Staphylococcus haemolyticus*, and *Enterococcus faecalis*, all bacteria isolated most frequently among patients of two previous studies [18,19]. The antimicrobial susceptibility profile assay of these strains reported multidrug-resistant patterns in all the microorganisms (Appendix A). The phylogroups and virulence genes of the *E. coli* strains were also analyzed in these assays, five strains were found to belong to the B2 group (O1, O6, O16, O25, and O75 serogroups), two to A (O9, O20), two to D (O7 and O17) and one to the B1phylogroup (O8). The presence of genes involved in virulence analyzed by PCR showed that strains from phylogroup B2 presented between 11 and 23 genes, the most frequent being *fimH*, *feoB*, *sitA*, *chuA*, and *ompA*, among others, while the strains from phylogroups A, B1, and D presented between 2 and 13 genes (Appendix A).

### 2.2. Procedures for Preparing Lysates

To determine the time in which the selected bacteria reached the exponential phase, growth kinetics were performed, in which it was identified that the enterobacteria report exponential growth between 3.5 and 4 h post-inoculation. In the case of Gram-positive bacteria, the exponential growth of *S*. *haemolyticus* starts at 8 h and *E. faecalis* 6 h post-inoculation. To have homogeneous parameters when adjusting the samples of the lysates under study, the concentrations of some of its components were analyzed. In the UNAM-HIMFG lysate, the concentration of proteins was 578 μg/mL, peptidoglycan 0.26 ng/mL, lipopolysaccharide 174 ng/mL, and carbohydrates 8657 μg/mL. When comparing the concentration of the same compounds in the monovalent lysates evaluated, it was observed that the UNAM-HIMFG polyvalent had concentrations up to five times higher than those obtained in the monovalent lysates (Table 1).

### 2.3. Antigenicity of UNAM-HIMFG and Monovalent Lysates

To confirm if the components of the UNAM-HIMFG lysate and some monovalent were immunogenic, SDS-PAGE and Western blot (WB) tests were performed. In the SDS-PAGE, the presence of fractions with different migration profiles was observed (Figure 1A,B). The antigenicity of the fractions obtained in the lysates was analyzed, using serum from a patient with CUTI who was treated with a monovalent lysate (autovaccine) prepared with an *E. coli* O25:H4 strain [18]. An intense serum reaction against the LPS of UNAM-HIMFG and monovalent *E. coli* O25:H4 lysates was observed, alongside poor recognition of the *E. coli* O20:H9 and *Citrobacter freundii* lysates (Figure 2). The result obtained shows that the autovaccine administered to a patient with CUTI activated the immune response, showing high specificity towards the LPS present in both lysates UNAM-HIMFG and O25:H4. Conversely, serum from patients with CUTI but who did not receive autovaccine did not react with either lysate.

### 2.4. Protein Detection

The SDS-PAGE performed shows fractions of different molecular weights (Figure 1), rabbit antibodies against OmpA previously obtained [18] were used in a WB assay. An intense reaction was observed on a 35 kDa (which corresponds to OmpA) and with less intensity against other fractions of different molecular weights (Figure 3). The result indicates that the UNAM-HIMFG lysate contains immunogenic protein fractions. In the assay analyzed, three different batches of the UNAM-HIMFG lysate showed a similar recognition profile.

### 2.5. Cytotoxic Activity of Lysates

The safety of the UNAM-HIMFG lysate and some monovalent lysates was analyzed on murine macrophages J774.A1 and the human epithelial cells Caco-2 and HEK293. The result of the three cell lines evaluated by trypan blue staining showed that none of the bacterial lysates caused damage to the challenged cells—an effect different from that induced by Triton X-100 and DMSO used as controls for cytotoxic effect. To confirm the safety of the lysates, MTT and XTT assays were performed on the HEK293 line. In this assay, it was observed that the lysates did not alter the viability of the cells as was observed in the untreated control; however, opposite results were observed in the cells treated with Triton X-100 and DMSO, where the cell damage was total (*p ≤* 0.005).

### 2.6. Effect of UNAM-HIMFG Lysate on a Mouse Model

A mouse model was used to determine whether the polyvalent lysate could cause damage to digestive tract tissues when administered orally at a level of 200 μL weekly for 60 days. The animals, during the time that the lysate was administered, did not show clinical alterations (weight loss, behavioral changes, signs of gastrointestinal alterations, or other alterations), suggestive of alterations or damage caused by the administered product. After 60 days, the animals were sacrificed and in the analysis of the stomach and intestinal sections, no alterations were identified, a situation similar to what was observed in the animals used as controls inoculated with physiological saline solution (Figure 4).

### 2.7. Immunostimulatory Activity of Lysates

The stimulatory capacity of UNAM-HIMFG and monovalent lysates of *E. coli* O25:H4 and O20:H9, *K. pneumoniae*, and *E. faecalis* was analyzed on the murine macrophage cell line J774.A1. The assays showed that all lysates except that of *E. faecalis* activated the production of TNF-α with a dose-dependent effect (Figure 5A–E). To find out whether the effect was due to LPS, polymyxin B neutralization was performed; in this test, a 38% decrease in TNF-α secretion was observed in the UNAM-HIMFG lysate, 45% in *E. coli* O25:H4, 26% in *E. coli* O20:H9, and 11% in the correspondent of *K. pneumoniae*, compared with the untreated samples (Figure 5A–E). In these cells, IL-10 production was also evaluated with negative results in all lysates at 24 h of incubation (*p* ≤ 0.005).

The activities of UNAM-HIMFG, *E. coli* O25:H4, and *E. faecalis* lysates were also analyzed on human macrophages. In this assay, it was identified that both UNAM-HIMFG and *E. coli* O25:H4 lysates activate TNF-α secretion, with no dose-dependent effect (Figure 6A,B), and again, *E. faecalis* lysate did not activate TNF-α secretion (Figure 6C). In this assay, the effect of polymyxin B on macrophage activation was also evaluated, this effect was 100% inhibited with UNAM-HIMFG lysate and 22% with *E. coli* O25:H4 lysate (Figure 6A,B). IL-6 production was also analyzed; the effect of both UNAM-HIMFG and *E. coli* O25:H4 was positive for the secretion of the cytokine (Figure 6D,E). As was previously observed, the stimulatory activity decreased by 50–60%, respectively (Figure 6D), after polymyxin B treatment. In a new finding, *E. faecalis* lysate showed no stimulatory activity (*p* ≤ 0.005) on IL-6 secretion (Figure 6F).

## 3. Discussion

The search for alternatives for the control of bacterial infections is of utmost importance considering the increasing number of multiresistant bacteria to antibiotics [1,5,6]. UTIs are infectious processes that have acquired great relevance, mainly because of the current difficulty in their effective treatment with antimicrobials [9,10,21]. Although the use of vaccines to prevent diseases is an alternative of great relevance, in many infectious diseases, the complexity of the pathogen or causal pathogens is a limiting factor that makes it difficult to obtain the optimal immunogen [22,23]. At the same time, since UTIs do not affect the entire population, the use of a prophylactic method to prevent its occurrence is complicated. Although UTIs are a condition of great clinical and epidemiological importance, it is not possible to practice generalized immunization [2,3].

This is why alternatives should be sought that contribute to the treatment and control of UTIs [24,25]. The use of bacteria lysed by different procedures complies with the above goals. Alrmroth E. Wright (1930) developed the first autovaccine when he considered that dead microorganisms, besides helping to prevent infections, could also contribute to their treatment [26]. In two previous studies, the effect of monovalent lysates was evaluated (autovaccines) in the treatment and control of recurrent urinary tract infections [18,19]. In the previously mentioned studies, *E. coli* of different serogroups was observed to be the most frequently isolated pathogen. With this information, the development of a polyvalent immunogen (bacterial lysate) was considered, including serogroups defined as “classic” uropathogenic *E. coli* (O25, O75, O1, O6, O8), mainly related to persistent infections, and other (O20, O11, O17), antigenic, variants that could be considered endemic, mainly associated with cases of reinfections. The immunogen also included *Klebsiella pneumoniae*, *Klebsiella aerogenes*, *Citrobacter freundii*, *Proteus mirabilis*, *Staphylococcus haemolyticus*, and *Enterococcus faecalis* bacteria, which, although less frequently, were also isolated from the studied patients. The polyvalent lysate UNAM-HIMFG, due to its composition, includes several molecules (Table 1) with immunogenic properties capable of activating the immune system [21,24,27,28,29]. This was confirmed by SDS-PAGE and WB analysis of the lysate. The composition (Figure 1) includes a great diversity of components of different molecular weights that are recognized by antibodies from patients with persistent UTIs treated with a monovalent lysate prepared with a UPEC strain of serogroup O25:H4. In this assay (Figure 2), LPS recognition was observed and found to have high specificity towards the homologous LPS (O25), but only in patients previously treated with the monovalent lysate [18,19], since serum from untreated patients did not induce a response.

The content and diversity of peptidoglycan and sugars, together with the LPS of the UNAM-HIMFG lysate (Table 1), support its broad immunostimulant capacity. There are studies that point out the participation of LPS in UTIs as a mediator in colonization events and the formation of intracellular reservoirs, as well as an immunomodulator in the response to these infections [30]. The immunomodulatory activity of LPS has been analyzed by making modifications in the molecule and evaluating the effect of the changes on the response of mice in the expression of cytokines and antibodies [30]. The UNAM-HIMFG lysate, in addition to having a high LPS content (Table 1), is diverse due to the different strains of enterobacteria that were selected for its composition, which confers greater complexity without making modifications to the molecule. As with LPS, the ability of different doses of bacterial polysaccharides and peptidoglycan to stimulate the systemic immune response in mice free of specific pathogens has been demonstrated [28].

It is important to note that outer membrane proteins are considered important epitopes for the development of diagnostic systems and mainly vaccines, because of their ability to stimulate the immune response [29,31,32,33]. The protein composition of the UNAM-HIMFG lysate is very diverse (Figure 1A,B); to know if some of these proteins were immunogenic, their reactivity was evaluated by WB assays. Since there are no antibodies against the lysate, rabbit polyclonal antibodies directed against OmpA obtained previously were used [18]. As mentioned above, the antibodies recognized a fraction of approximately 35 kD corresponding to OmpA (Figure 3), as well as other (unidentified) protein fractions. This result was like that obtained previously when analyzing different monovalent lysates prepared with *E. coli* strains of serogroups O25, O75, O8, and O6 [18]. It is important to mention that in the previous study, in addition to OmpA, OmpC, NmpC, Udp, Gpma, FimH, BamB, and BamC were identified. The presence of the referred protein fractions is probably part of the pathogen-associated molecular patterns (PAMPs), which participate in the immune stimulation of the host against uropathogens [25,34,35,36].

The prepared bacterial lysate was obtained by heat treatment with saturated steam and then subjected to filtration, so it is a compound free of whole bacterial cells but retains most of the soluble components including those present in the periplasmic space, which may have different properties. NlpD is a protein associated with bacterial cell division that activates the peptidoglycan hydrolase; previous studies state that this protein phosphorylates host RNA polymerase II and by transcriptional regulation, decreases the inflammatory response secondary to UPEC infection so that a severe acute infection is reversed and becomes asymptomatic bacteriuria [37,38,39]. Although to date, the presence of this protein in UNAM-HIMFG lysate has not been identified, in previous studies in which autologous lysates were administered, some patients reported clinical improvement, although bacteriuria persisted [18,19].

The safety of a compound must be evaluated to avoid adverse effects on the individual to whom it is intended to be administered. As previously mentioned, UNAM-HIMFG lysate contains LPS in high concentrations; this compound has different effects that are concentration- and host-dependent. It is known that at doses of 10–15 μg/kg, it induces toxic shock in humans but not in mice, which tolerate several times this dose (10 mg/kg). In some studies, an adverse effect has not been demonstrated at doses of 20 µg/mL orally administrated for 10 consecutive days [40,41]. In this regard, it has been shown that the pure toxic fraction of LPS does not have the ability to cross the intestinal barrier and can only cause toxic effects when the mucosa undergoes a continuity solution [42]. LPS as well as any drug administered orally will present modifications during its passage through the digestive tract, from its entry through the mouth until its arrival in the intestine [43]. In this regard, it is known that treating LPS with acid reduces its toxic capacity, although it retains its stimulant properties [44]. In vitro and in vivo assays (Figure 4) to evaluate whether the UNAM-HIMFG lysate was innocuous showed that it is a non-toxic and safe compound despite its high content of LPS and peptidoglycan.

The results obtained in the safety assays of the UNAM-HIMFG lysate provided a guideline to evaluating its effect on cells associated with the immune response. The assay to evaluate the immunostimulatory activity of the lysates was performed on murine cells (J774A.1) in culture and on human macrophages [45,46,47]. UNAM-HIMFG as well as some of the monovalent lysate tested (*E. coli* O25H:4, *E. coli* O20:H9, and *Klebsiella pneumoniae*), activated the innate immune response in mouse and/or human macrophages by stimulating TNF-α production (Figure 5A,B). The results confirm that LPS is a key molecule for the activation of the innate immune response. The synergistic effect between LPS and bacterial DNA to induce TNF-α production in mouse macrophages has been described [48,49,50,51]. PG and lipoteichoic acids in human monocytes also have been stated to induce the secretion of this cytokine [50]. Although the monovalent lysate of *E. faecalis* did not show a stimulatory effect on either of the two cell types used, the result confirms that bacterial structures vary between genera and species [52,53], which supports the relevance of using immunogens with epitopes expressed by the different microorganisms [24,30,32]. In addition, with regard to LPS, it has been reported that in conjunction with bacterial flagellin, it induces TNF-α production [54]. It is important to note that the presence of flagellin in the UNAM- HIMFG lysate was because the *E. coli* strains used had flagellar antigens [10].

The role of TNF-α in urinary tract infections is of great relevance, in this respect, it has been observed in patients with rheumatoid arthritis treated with specific TNF-α inhibitors that the risk of suffering UTIs is notably increased [55]. The ability of the lysates (polyvalent and monovalent) to induce the expression of TNF-α suggests their participation in cell training to activate the immune response against pathogens responsible for UTIs [10,18,19].

In different studies in which cystitis images were evaluated, the participation of resident macrophages in the bladder that release TNF-α and induce the expression of chemokine [C-X-C motif] ligand 2 (CXCL2) was observed, as well as the activation of neutrophils that stimulates the expression of matrix metalloproteinase-9, which increases its capacity to cross the epithelium favorably modifying cystitis problems [56,57]. Finally, the resident mast cells of the bladder by secreting TNF-α generate the recruitment of dendritic cells [58]. The above shows the relevance of TNF-α in recurrent UTIs and how its activation by the effect of a polyvalent immunogen can participate in the treatment and control of UTIs by limiting the use of antimicrobials.

The study shows the ability of monovalent and polyvalent lysates to induce in mouse cells and human macrophages the production of TNF-α, but not IL-10. Studies with human peripheral blood monocytes indicate that the presence of neutrophils in an apoptotic state and urinary epithelial cells is required for IL-10 release. What was observed in this study could be explained by the fact that the information was obtained from a cell line [J774.A1] and cultured human macrophages [59,60,61]. The study analyzed the presence of different genes related to *E. coli* virulence; the result was important because of the number of genes presented by the selected strains (Appendix A), some of which were related to the expression of toxins, fimbriae, and other structures relevant to *E. coli* virulence. The study also showed that UNAM-HIM lysates and monovalent *E. coli* O25.H4 activated IL-6 production (Figure 6A,B). It has been found that the different components of UPEC strains (type 1 fimbriae, curli, and flagellum) in co-culture systems participate in the activation of IL-6 production [61]. The activation of TNF-α and IL-6 production suggests that the response activated by the components of the lysates is mainly pro-inflammatory and directed towards cell capacitation and migration. However, it is important to note that the lysates were evaluated only with macrophages, so the immunological context of how the lysates act on other cytokines and different cell signaling pathways needs to be further understood.

In animal models, it has been demonstrated that the capacity of LPS administered orally and intranasally to conduct a humoral response mainly by IgA, the immunoglobulin is important in the response for the neutralization of urinary pathogens during infection [62,63,64]. This work shows the immune recognition of LPS by systemic antibodies and the activation of pro-inflammatory cytokines, which shows the relevance of this molecule with great structural diversity to the different strains of *E. coli* that have the UNAM-HIMFG lysate, which, as has been reported, is relevant to the modulatory response of the host infection [22,30,48,52,63]. There are different commercial formulations prepared from bacterial lysates, in which some components’ presence needs to be analyzed in the UNAM-HIMFG lysate [65,66].

## 4. Materials and Methods

### 4.1. Ethics Agreements

All experiments were conducted in accordance with the specific techniques for the production, care, and use of laboratory animals described in NOM-062-ZOO-1999 [67] and have been approved by the Institutional Subcommittee for the Care and Use of Experimental Animals (SICUAE) of the Faculty of Veterinary Medicine and Zootechnics of the National Autonomous University of Mexico (FMVZ-UNAM) [Protocol No. SICUAE.DC2020/3-1], the Internal Committee for the Care and Use of Laboratory Animals (CICUAL) of the Faculty of Medicine of the National Autonomous University of Mexico (FacMed-UNAM) [Protocol No. CICUAL 002-CIC.2021], and the Research, Research Ethics, and Biosafety Committees of the Hospital Infantil de México Federico Gómez [HIM-2023-008].

### 4.2. Microorganisms

The UNAM-HIMFG lysate was prepared with 10 strains of *E. coli* of different serotypes and one bacterial strain each of Klebsiella pneumoniae, Klebsiella aerogenes, Citrobacter freundii, Proteus mirabilis, Staphylococcus haemolyticus, and Enterococcus faecalis isolated in two previous studies [10,18]. The selected preserved strains were recovered by culture on Luria-Bertani (LB), MacConkey (McC) agar (Bioxon) for enterobacteria, Mannitol agar with Salt (Mcdlab) for Staphylococcus, and Bilis agar with esculin and sodium azide (Merk) for Enterococcus. The identities of the strains were confirmed by verifying their biochemical profile [68,69]. The recovered strains were preserved in skim milk (15%) and glycerol (20%) and in LB broth with glycerol (20%), and maintained at −70 °C until use. Once their identities were confirmed by metabolic tests, the E. coli strains were serotyped using 186 anti-O and 56 anti-H sera (SERUNAM, Mexico) obtained from rabbits immunized with the respective antigens (somatic [O] and flagellar [H]); the microplate agglutination procedure (Orskov) was used for this assay [70].

#### 4.2.1. Virulence Genes and Phylogroups of *Escherichia coli*

To confirm the UPEC gene profile of the selected *Escherichia coli* strains, Polymerase Chain Reaction (PCR) assays were performed. DNA extraction was performed by the guanidine thiocyanate procedure described by Pitcher et al. [71], with some modifications [72]. Briefly: The obtained DNA was reconstituted with 50 μL of MiliQ water (Millipore) and stored at −20 °C until use, and DNA concentration was adjusted to 100 ng/μL. PCR assay for virulence genes was performed with the gene-specific primers used in previous studies [10,18,19] and others used as a complement (Appendix A) [73,74,75]. Assays were performed with PCR Master Mix (2×) (Thermo Fisher Scientific, Waltham, MA, USA) with 0.2 nM of each primer and 200 ng of DNA; for phylogroups, the procedure reported by Clermont [76] was used. Each tube with its mixture was placed on the plate of a thermal cycler (Techne 3Prime, Thermo Scientific Fisher, Waltham, MA, USA; MiniAmp Thermal Cycler, Applied Biosystems, Thermo Scientific Fisher, Waltham, MA, USA) following the conditions referred to in Appendix A. The amplified genes were analyzed in 1.5% agarose gel, performing electrophoresis in sodium borate (SB) buffer at 60 volts for 60 min or 100 volts for 30 min [72]. The following were used as reference strains: *E. coli* UPEC CFT073 (ATCC 700928), *E. coli* K12 HB101 (ATCC 33694), and poultry *E. coli* APEC 1331 and APEC 1336 (Appendix A).

#### 4.2.2. Antimicrobial Sensitivity

All the microorganisms used to elaborate the UNAM.HIMFG, were tested for antimicrobial sensitivity by diffusion on Müeller–Hinton agar (BD Bioxon, Ciudad de México, Mexico) with sensidiscs (Oxoid, Sigma-Aldrich, St. Louis, MO, USA) impregnated with the following antimicrobials: ampicillin (10 μg), mecilinam (10 μg), amoxicillin with clavulanic acid (20 μg/10 μg), piperacillin with tazobactam (100 µg/10 μg), cefozolin (30 μg), cefamandole (30 μg), cefepime (30 μg), cefoperazone (75 μg), cefoxitin (30 μg), ceftriaxone (30 μg), ceftazidime (30 μg), cefuroxime (30 μg), meropenem (10 μg), nitrofurantoin (300 μg), aztreonam (30 μg) (except Gram-positive), gentamicin (10 μg), amikacin (30 μg), kanamycin (30 μg), trobramycin (10 μg), streptomycin (10 μg), tetracycline (30 μg), ofloxacin (5 μg), ciprofloxacin (5 μ), norfloxacin 10 (μg), nalidixic acid (30 μg), trimethoprim with sulfamethoxazole (1.25 μg/23.75 μg), sulfonamides (250 μg), trimethoprim (5 μg), chloramphenicol (30 μg), and fosfomycin (200 μg); in addition to vancomycin (30 μg) in the case of Gram-positive for use in urinary tract infections as recommended by CLSI [77]. *E. coli* strain ATCC 25,922 was used as a control for the assay.

### 4.3. UNAM-HIMFG Lysate

#### Growth Curves of Selected Bacteria

To obtain the growth profiles of the different microorganisms that would integrate the immunogen, bacterial growth kinetics were performed. Bacteria were individually cultured on LB agar incubated 24 h at 37 °C ± 2 °C. From the development of the culture, 5 colonies were selected and inoculated in 7.5 mL of LB broth incubating at 37 °C ± 2 °C for 16 h under agitation (200 rpm). From these cultures, 2 μL were taken and inoculated in 200 μL of LB broth placed in a 96-well plate (Costar, Corning, Austin, TX, USA) to analyze the growth kinetics for 8 h incubating the plate at 37 °C without agitation with readings (O.D. 600 nm) every hour in a spectrophotometer (Epoch BioTeK, Agilent Technologies, Santa Clara, CA, USA). Uninoculated culture medium was used as a blank performing the same procedure. With the growth curve data of the selected bacteria, the polyvalent lysate was prepared using the procedure previously described [18], with some modifications. Briefly: with each bacterium, a pre-inoculum was prepared from 5 colonies in 7.5 mL of LB broth to optimize the adaptation of the bacteria to the liquid medium; then, they were transferred to 200 mL of LB incubating 37 °C ± 2 °C in constant agitation at 200 rpm in constant agitation at 200 rpm until reaching the exponential phase of growth. The bacterial mass was obtained by centrifugation at 4000× *g* for 60 min and the supernatant was discarded to subsequently perform three washes suspending the bacteria package each time in 30 mL of physiological saline solution (PSS) (Pisa, CdMx, Ciudad de México, Mexico). Once the supernatant of the third wash was eliminated, the pellet was reconstituted with 25 mL of SSF, and the bacterial suspension was adjusted to a value of 0.582 at an optical density (O.D.) of 600 nm, equivalent to approximately 9 × 10^8^ CFU/mL. The referred procedure was performed for all the selected bacteria, with the defined concentration, and the microorganisms were mixed in 100 mL of SSF. To inactivate the bacteria, the flask with the mixture was placed under saturated steam (110 °C) for one hour maintaining the pressure at 6.4 psi (0.45 kg/cm^2^). Once the bacterial mixture was inactivated, the flask was centrifuged (4000 g/1 h) to obtain the supernatant, which was subsequently filtered with nitrocellulose or polyvinylidene fluoride (PVDF) membranes of 0.22 μm diameter (Merk Millipore Darmstadt, Germany). In parallel, monovalent lysates of *E. coli* serogroups O25:H4 and O20:H9, *K. pneumoniae*, and *E. faecalis* were prepared with the same procedure as previously mentioned. Each of the sterile lysates was packaged under aseptic conditions in 5 mL vials, previously baked (170 °C/2 h.) and autoclaved. The vials with the lysates were subjected to sterility tests and finally stored at 4 °C until use.

### 4.4. Polyvalent Lysate Components

The concentration of some of the components of the UNAM-HIMFG lysate and the monovalent *E. coli* serogroups O25:H4 and O20:H9, *K. pneumoniae*, and *E. faecalis* were evaluated. Proteins of the compounds were precipitated with trichloroacetic acid (TCA) following the indications previously described [78] with the following modifications: to 500 L of bacterial lysate was added one volume of 20% TCA; this was incubated on ice for 20 min, centrifuged at 13,000× *g*/20 min to remove the supernatant, and then the pellet with protein was washed with 3 volumes of cold acetone, again centrifuged (11,000× *g*/5 min); then, the supernatant was decanted and the pellet was dried (37 °C/10 min); and this was then reconstituted with 500 μL of physiological saline with 0.01 N NaOH. Protein concentration was evaluated by the Bradford colorimetric method (Bio-Rad Protein Assay) following the manufacturerʹs instructions, taking readings at a wavelength of 595 nm. The pH of the immunogens was evaluated with test strips (Hydrion) by placing 10 μL of the product on the strip. The peptidoglycan and lipopolysaccharide concentration of the lysates was analyzed by enzyme immunoassay with ELISA kit (Abbexa LCC; Houston, TX, USA), considering the manufacturerʹs instructions. For carbohydrate concentration, the phenol-sulfuric acid colorimetric method [79] was used; for this purpose, the lysates samples were adjusted to 200 μg and 5 μL of 80% phenol and 500 μL of pure H_2_SO_4_ were (carefully) added. Samples were incubated for 20 min at 30 °C, followed by spectrophotometer readings at an absorbance of 490 nm [80]. All assays were performed three times in duplicate.

#### Lysates SDS-PAGE Analysis

The presence of the components of the lysates (UNAM-HIMFG and monovalent) was analyzed by electrophoresis in 12% polyacrylamide denaturing gels with sodium dodecyl sulfate (SDS-PAGE) using 25 mM Tris base, 192 mM glycine, and 0.1% SDS as running buffer; 100 volts were applied to the samples [80]. In each well of the gel, a volume with an approximate concentration of 20 μg/mL of protein was placed; the protein profiles of the gel were analyzed by staining with silver nitrate as previously described [81].

### 4.5. Lysates Immunogenicity

To determine whether the serum of patients immunized with monovalent autovaccine [10,18] recognized any of the protein fractions of the lysate observed in SDS-PAGE, a Western blot assay was performed. For this, new 15% polyacrylamide gels were prepared, and the separated fractions were transferred to nitrocellulose membranes (Amersham Protran). The membrane was saturated with 5% skim milk and incubated for 30 min, then the membranes were washed with phosphate buffer solution (PBS) + Tween. Serum (1:50) from a patient treated with autologous immunogen prepared with an *E. coli* O25 strain was added to each membrane. Serum from individuals with UTIs but not treated with immunogen was used as a negative control, and rabbit anti-human IgG (1:1500) labeled with rabbit peroxidase was used as a secondary antibody. To determine whether the UNAM-HIMFG lysate contained OmpA, SDS-PAGE and Western blot were performed on three batches of the lysate; rabbit antibodies (1:250) immunostained with a partially purified fraction of OmpA were used to identify the protein; goat anti-rabbit IgG (1:1500), labeled with horseradish peroxidase as previously described [18] was used as a secondary antibody to visualize the reaction.

### 4.6. Cytotoxic Activity of Lysates Evaluated with Trypan Blue

For this assay, murine-derived macrophage lines J774A.1 ATCC TIB-67, kidney epithelial cells of human origin [HEK-293 ATCC CRL-1573], and human colon epithelial cells [Caco-2 ATCC HTB-37] cultured in RPMI medium (Sigma-Aldrich, St. Louis, MO, USA) supplemented with 10% fetal bovine serum, 2 mM glutamine, 1 mM sodium pyruvate, penicillin- streptomycin-gentamycin-amphotericin B (100UI/100 μg/100/2 μg/mL), incubated at 37 °C with 5% CO_2_. Approximately 3 × 10^5^ cells of each line were plated in 24-well plates, and to each well, 100 ng/mL protein lysate (UNAM-HIMFG, *E. coli* O25, *K. pneumoniae*, *E. faecalis*) was added. In addition, the effect of *Salmonella* Enteritidis Lipopolysaccharide (Sigma-Aldrich, St. Louis, MI, USA) was evaluated at concentrations of 100 ng/mL and 1 μg/mL; dimethyl sulfoxide (DMSO) [1.5 M] was used as a positive control and RPMI medium as a negative control. The plates were incubated at 37 °C for 24 h with 5% CO_2_. At the end of the incubation time, the cells were disaggregated with 4 mM PBS-EDTA or Trypsin EDTA (Sigma-Aldrich, St. Louis, MI, USA) for 10 min at 37 °C with 5% CO_2_. The disaggregated cells were stained with 0.4% Trypan Blue (Sigma-Aldrich, St. Louis, MI, USA) and subsequently observed and quantified in a Neubauer chamber by analyzing the cytopathic effect (vacuolization, syncytia formation), or death by viable cell count [82]. The assay was evaluated by three different observers and the following score was assigned in terms of survival or damage observed 0: 85–100% (no damage), 1: 26–84% (damage), 2: 11–25%, and 3: 0–10% (cytotoxic).

#### Cytotoxicity Activity Evaluated with MTT and XTT

A three-passage culture with HEK-293 cells was brought to 85–95% growth confluence and then the cells were detached from the culture bottle using PBS and Trypsin + EDTA (0.25%) (Sigma-Aldrich, St. Louis, MI, USA) for 5 min, collected and centrifuged at 300× *g* for 5 min at 25 °C. Approximately 50,000 cells × mL per well were plated in 96-well plates and incubated for 96 h at 37 °C with 5% CO_2_. In each well, 100 ng/mL of protein from each of the lysates was independently spotted following the sequence referred to previously (4.6) including an additional positive control with Triton 100× (0.01% *v*/*v*). As in the previous assay, the cells were incubated 24 h at 37 °C with 5% CO_2_, and at the end of the incubation time, the cells were washed twice with SSF (37 °C). The MTT (3-(4,5-dimethylthiazol-2-yl)-2,5-diphenyltetrazolium bromide) (Roche, Basel, Switzerland) and XTT (2,3-bis-(2-methoxy-4-nitro-5-sulfophenyl)-2H-tetrazolium-5-carboxanilide) (Roche, Basel, Switzerland) assay was performed following the manufacturerʹs instructions; the incubation with the reagents was performed for 24 h. At the end of the process, readings were taken at optical densities of 500 nm and 600 nm for MTT and XTT, respectively. Taking as reference the absorbance of the cells without stimulus as 100% of viability, the assay was performed 3 times in duplicate [83].

### 4.7. In Vivo Effect of UNAM-HIMFG Lysate

Six female mice from six to eight weeks old of the Balb/c strain, bred and housed in the biotherium unit of the Hospital Infantil de México Federico Gómez (HIMFG), were used. The animals were maintained in sterile rack conditions with 20 airflow changes per hour at a relative humidity of 45 to 65% and a temperature of 18–22 °C. Each group of animals was orally administered 200 μL of UNAM-HIMFG or SSF lysate every week with a 22G stainless steel rigid gastric tube. The animals were kept for 60 days and at the end of the period, were euthanized (after anesthesia with xylazine in association with ketamine 100 mg/kg and 5 mg/kg, respectively) by cervical dislocation according to NOM-062—ZOO-1999, [67].

Stomach and small and large intestines were collected for histological evaluation of potential damage induced by the immunogen in the gastrointestinal mucosa. For tissue collection, the cardial portion of the stomach and the rectal portion were ligated with cotton thread and then buffered with formaldehyde at pH 7.0 introduced into the digestive lumen using a 10 mL syringe with a 24 G needle. Once the tissues were fixed, the organs were placed in a flask with buffered formalin at pH 7.0 for histopathology studies [84]. The fixed tissues were fractionated and embedded in kerosene for sections and Hematoxylin and Eosin (H and E) staining.

### 4.8. Stimulatory Activity of UNAM-HIMFG and Monovalent Lysates in Mouse Macrophage Cell Lines and Primary Human Macrophage Cultures

The murine macrophage cell line J774A.1 and monocytes obtained from human peripheral blood leukocytes were used [85]. Human mononuclear cells were obtained from bags of leukocyte concentrates from samples from four donors following the approval guidelines established by the Bioethics Committee of the blood bank of the HIMFG. The mononuclear cell fraction was obtained by Lymphopred gradient centrifugation (Nycomed AS, Oslo, Norway). Once the cells were separated, they were washed three times with physiological saline and the obtained monocytes were purified by the negative selection method using a mixture of anti-CD3, CD7, CD19, CD45RA, CD56, and anti-IgE antibodies (Miltenyi Biotec, Bergisch Gladbach, Germany). Purified monocytes (1 × 10^5^ monocytes/mL) were differentiated into macrophages by culturing the cells for 7 days in 24-well plates (Costar, Corning, Austin, TX, USA).

Murine macrophages J774A.1 were cultured in RPMI medium (Sigma-Aldrich, St. Louis, MI, USA) supplemented with 10% fetal bovine serum, 2 mM glutamine, 1 mM sodium pyruvate, and penicillin-streptomycin-gentamicin-amphotericin B (100UI/100 μg/100/2 μg/mL) until 85–95% confluence was obtained. The cell monolayer was disaggregated with phosphate buffered solution (PBS) pH 7.0–7.2 with 4 mM diethylene aminotetraacetic acid (EDTA) and 2 × 10^5^ cells/mL was used for stimulation. Plates with human and murine J7741.A macrophages were challenged with decreasing protein concentrations of 100 ng/mL, 10 ng/mL, and 1 ng/mL of the lysates; UNAM-HIMFG, *E. coli* O25:H4, *E. coli* O20:H9, and *E. faecalis*, *Salmonella* Enteritidis Lipopolysaccharide (Sigma-Aldrich, St. Louis, MO, USA) 100 ng/mL, Polymyxin B and polymyxin B (Sigma-Aldrich, St. Louis, MO, USA)-treated stimuli [100 ng/mL stimulus + 100 ng/mL Polymyxin B] were used as controls. Plates were maintained for 1 h at room temperature. Once the cells were stimulated with the extracts, they were incubated for 24 h at 37 °C with 5% CO atmosphere_2,_ after stimulation the supernatant was harvested and stored at −70 °C until use.

The expression of TNF-α, IL6, and IL10 was performed by enzyme immunoassay with the commercial ELISA Ready-SET-Go Kits (Thermo Scientific Fisher, Waltham, MA, USA) and HUMAN ELISA Set (Becton-Dickinson, East Rutherford, NJ, USA), following the manufacturerʹs instructions an extrapolation of cytokine concentrations was performed according to a recombinant cytokine standard curve for each of the samples evaluated in duplicate.

### 4.9. Statistical Analysis

ANOVA tests with Tukeyʹs post hoc were performed using the R studio platform version 8.1 and GraphPad Prism v10.1.1.

## 5. Conclusions

The UNAM-HIMFG lysate is composed of bacterial strains isolated more frequently in two prospective studies of recurrent UTIs. The antigenic diversity of the lysate favors its immunogenic activity and likely its positive effect in the treatment and control of CUTI. The effect it induces is likely associated with that reported for NlpD, which inhibits the host’s RNA polymerase II (Pol II), which inhibits the expression of pro-inflammatory genes, favoring the reduction of symptoms. However, to confirm the above, it is necessary to identify the presence of NlpD in the lysate and to evaluate the effect of the UNAM-HIMFG lysate in an animal model. Although the use of bacterial lysates is an alternative for controlling UTIs without the use of antimicrobials, there are other alternatives that are being explored. Among these are the development of recombinant vaccines, the use of phage therapy, probiotics, postbiotics, and even the search for new antimicrobials or the use of natural products; all with the purpose of treating and controlling UTIs and avoiding their consequences, as well as reducing the use of antimicrobials as much as possible.

## Figures and Tables

**Figure 1 ijms-25-06157-f001:**
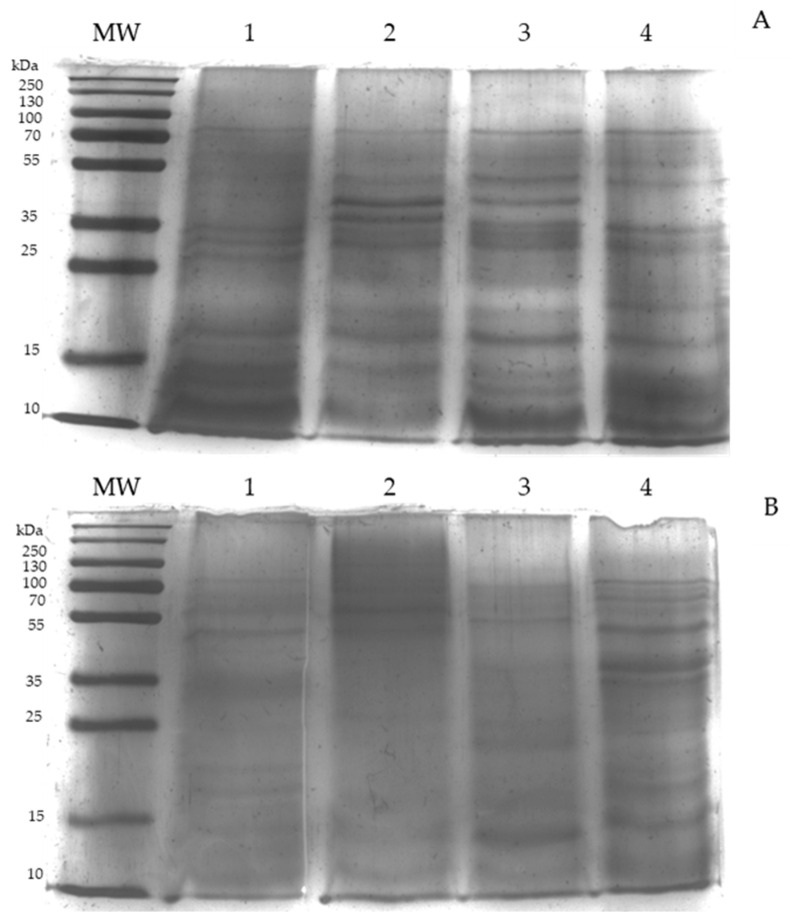
SDS PAGE of polyvalent and monovalent lysate components Silver stained. (**A**): Lines: (1) Polyvalent lysate (UNAM-HIMFG), (2) *E.coli* O25, (3) *E.coli* O20, (4) *Citrobacter*, (**B**): Lines: (1) *Klebsiella*, (2) *Enterococcus*, (3) *Staphylococcus*, and (4) *Proteus*. *MW Molecular Weight: PageRuler Plus Prestained Protein Ladder* (ThermoScientific).

**Figure 2 ijms-25-06157-f002:**
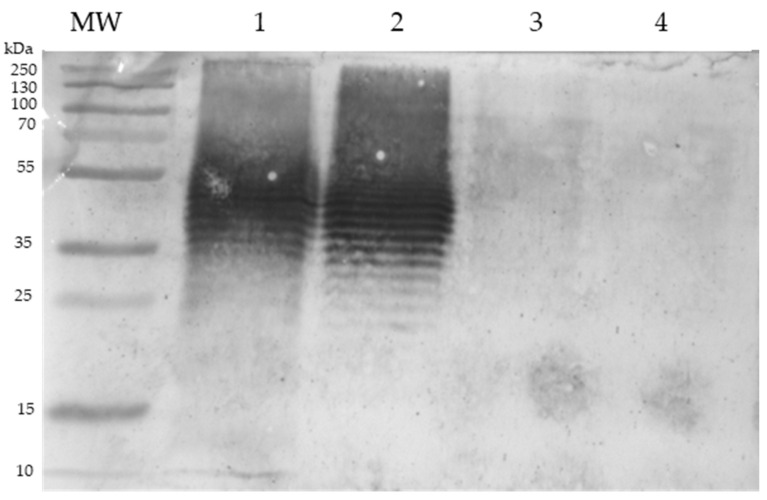
Western blot of polyvalent (UNAM-HIMFG) and monovalent lysates. The lysates were challenged with a serum sample obtained from a patient previously treated with a lysate prepared with an O25 *E. coli* strain. Lines: (1) UNAM-HIMFG; (2) *E.coli* O25; (3) *E.coli* O20; and (4) *Citrobacter*. *MW Molecular Weight*: *PageRuler Plus Prestained Protein Ladder*.

**Figure 3 ijms-25-06157-f003:**
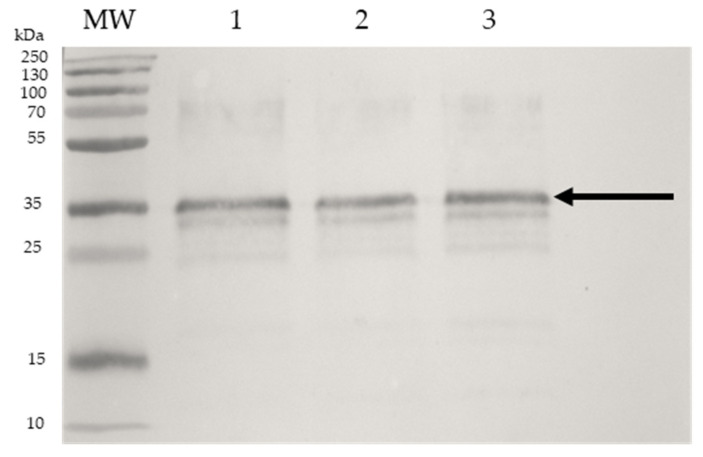
Western blot of different preparations of the UNAM-HIMFG polyvalent lysate. The WB was developed with polyvalent rabbit antibodies obtained against OmpA. Lines: 1, 2, and 3, samples of different preparations of polyvalent lysate. (Arrow indicates OmpA 35 kDa) *MW Molecular Weight*: Page Ruler Plus Protein Ladder pre-dye.

**Figure 4 ijms-25-06157-f004:**
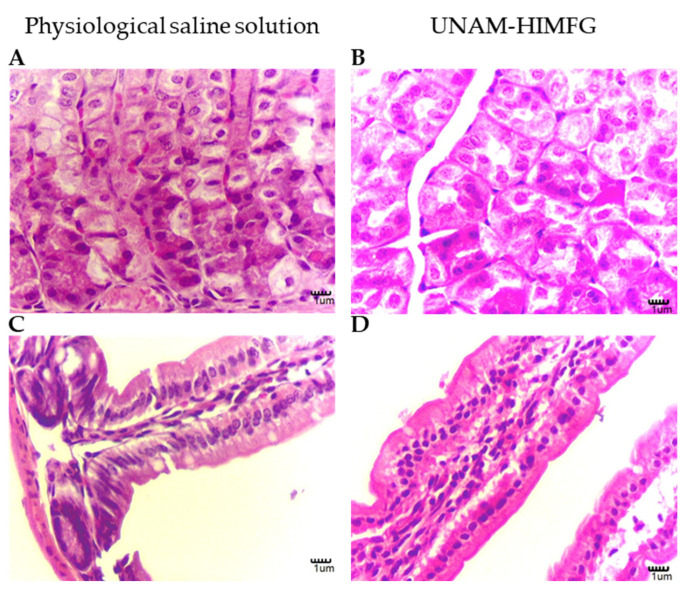
Histological evaluation of mice tissues treated with physiological saline solution (**left**) or the polyvalent bacterial lysate (**right**). (**A**,**B**) Stomach epithelium, (**C**,**D**) Intestinal epithelium. Samples were stained with Eosin and Hematoxylin and observed to ×400.

**Figure 5 ijms-25-06157-f005:**
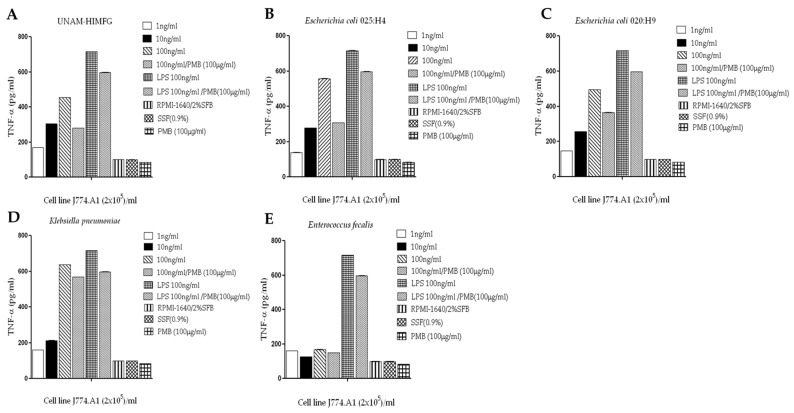
Secretion of TNF-α in J774 A.1 murine macrophages with different concentrations of the lysates. (**A**) UNAM-HIMFG, (**B**) *E. coli* O 25:H4, (**C**) *E. coli* O20:H9, (**D**) *K. pneumoniae*, and (**E**) *E. faecalis*. Results are the average of three tests performed in duplicate *p* ≤ 0.005. SSF: Physiological saline solution; PMB: polymyxin B 100 μg/μL; LPS: Lipopolysaccharide 100 ng/μL.

**Figure 6 ijms-25-06157-f006:**
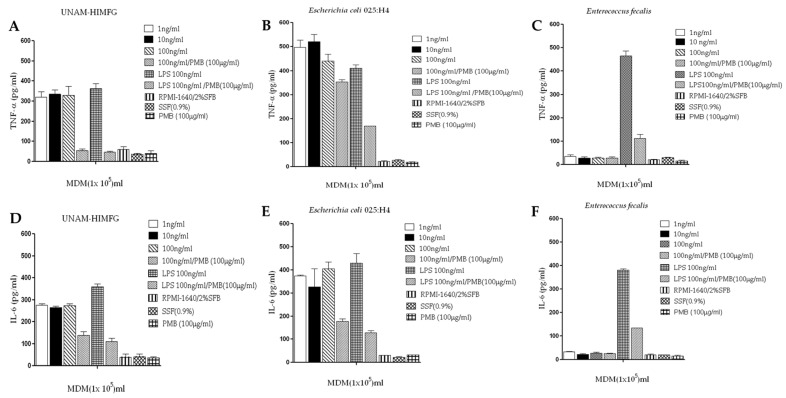
TNF-α and IL-6 production of human macrophages activated by UNAM-HIMFG, *E. coli* O 25:H4, and *E. faecalis* lysates. UNAM-HIMFG (**A**,**D**), *E. coli* O 25:H4 (**B**,**E**), y *E. faecalis*. (**C**,**F**). Results are the average of three tests performed in duplicate. *p* ≤ 0.005. SSF: Physiological saline solution; PMB: polymyxin B 100 μg/μL; LPS: Lipopolysaccharide 100 ng/μL.

**Table 1 ijms-25-06157-t001:** Analysis of macromolecules composing the different lysates.

Lysate	Protein Concentration (μg/mL)	Peptidoglycan Concentration (ng/mL)	Lipopolysaccharide Concentration (ng/mL)	Carbohydrate Concentration (Hexoses) (μg/mL)	Carbohydrate Concentration (Pentoses) (μg/mL)
UNAM-HIMFG	578	0.26	174	7823	8657
*Escherichia coli* O25:H4	25.8	0.01	1.2	500	569
*Escherichia coli* O20:H9	33.8	ND	ND	ND	ND
*Klebsiella pneumoniae*	23	0.033	1.4	1036	835
*Citrobacter freundii*	37.3	ND	ND	ND	ND
*Proteus mirabilis*	38	ND	ND	ND	ND
*Enterococcus faecalis*	33.1	0.017	0	409	377
*Staphylococcus haemolyticus*	20.8	0.08	ND	ND	ND

Abbreviation: ND, No Determinated. Values shown are averages obtained from at least two assays at different times; The pH in both monovalent and polyvalent lysates was 4.

## Data Availability

Not applicable. All data regarding this research are available in the present manuscript and no data was included in any database elsewhere.

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
