# Peer review of "Polyvalent Bacterial Lysate with Potential Use to Treatment and Control of Recurrent Urinary Tract Infections"

_ijms, 2024, doi:10.3390/ijms25116157_

Round 1

Reviewer 1 Report

Comments and Suggestions for Authors

The authors present a novel experimental study on the potential use of polyvalent bacterial lysates to treat and control recurrent UTIs. They found that the lysates demonstrated safety in vitro and in the animal model. It also appeared to activate the immune response. This is a well-designed study on a very clinically relevant topic. The results are impressive, and the manuscript is well written. Please elaborate on the next research steps. Does this study provide enough support to conduct an experimental study on the use of bacterial lysates for UTI control?

Author Response

Responses to comments issued by reviewer 1

We appreciate your valuable comments on our work, and we will send a response to your question, which we hope includes the requested information.

 Question:

 Please explain the next steps of the investigation. Does this study provide enough support to conduct an experimental study on the use of bacterial lysates for the control of urinary tract infections?

Answer:

Previously, we carried out a prospective study in children and adults to know the main bacteria related to recurrent urinary tract infections. In this same work, the efficiency of bacterial lysates (called “autovaccines”) was evaluated, for the treatment and control of chronic infections. The results were satisfactory in 70% of both groups of patients, in which the responsible microorganism was eliminated, and reinfection was controlled for periods of between seven months and up to a year or more.

The purpose of the evaluated manuscript was to select the more common microorganisms identified during both studies and to prepare a polyvalent immunogen. The bacteria were inactivated by heat and subsequently filtered to ensure the safety of the compound, and with this, the tests presented in the manuscript were carried out.

To corroborate whether it has a behavior like that identified in the lysates of autologous bacteria (autovaccines), a mouse infection model has been implemented, the results of which are in the analysis and writing stage for submission and potential publication.

Reviewer 2 Report

Comments and Suggestions for Authors

Authors provided a paper entitled “Polyvalent bacterial lysate with potential use to treatment and control of recurrent urinary tract infections” for the publication in IJMS.

This paper has a quite large scientific soundness, since it concerns the overuse of antibiotics, that is generating a lot of bias in patients, that tend to use them as soon as they have a minimum infection.

 The use of English is not brilliant, and requires to be revised accordingly.

Here is the list of my issues:

Line 47. “Usefulness” please consider a substitution for this word.

Line 74. “Treatment of UTIs is mainly with antimicrobials” improve syntax here.

Line 82. “to host organs with consequent repercussions” please define which kind of repercussion could be caused, especially patients outcomes.

Line 86. please define clearly ʺautovaccineʺ

Line 99 “previously isolated bacteria [12,13], which were 99 selected considering the frequency with which they were isolated” please check the use of English, avoiding repetitions.

Moreover, I would suggest expanding the sentence concerning the goals of this study, clarifying side aspects.

Line 112. “strains” should this be in italics, as well?

Line 121. the sentence seems to end in this manner. Please, remove the space here after “between” and link with the following part of the sentence.

Line 126. “578.22 μg/mL” I suggest approximation to unity.

Line 126. “peptidoglycan 0.2599 ng/mL” I suggest an approximation to the first, max second decimal unit, since we are talking about nanograms.

Same observations for the following two decimal digits.

Table 1 contains pH equal to 4 in all the compositions of lysate. This could be avoided by simply write it in the manuscript text presenting the table.

Line 135. “To find out” I would substitute with “to define/determine”

Line 221. “In the analysis” please check the syntax and the use of English.

Concerning the discussion section, besides discussing the therapeutic approach for UTIs, consider mentioning other potential strategies, such as vaccine development, new drug delivery systems for vaccine delivery (liposomes, niosomes, foams) or alternative therapies like phage therapy. Discuss how finding alternative treatments for UTIs could improve patient outcomes and reduce healthcare costs.

Line 320. “studies that point out the” references are needed here.

Line 363. “20 𝝻g/mL” the “𝝻” should not be bold.

Line 365. “are due to the change of pH and it has been demonstrated that LPS treated with acid” too long sentence; please divide it.

Conclusions need to be expanded, maybe moving some paragraphs or sentences from the very long discussion section. The definition of further perspectives could be also well appreciated.

Comments on the Quality of English Language

Authors need to revise their paper: use of English is poor.

Author Response

Responses to comments issued by reviewer 2

We appreciate the feedback on our work, we hope we have correctly interpreted what was suggested.

Line 47. “Usefulness” please consider a substitution for this word.

The change was made.

Line 74. “Treatment of UTIs is mainly with antimicrobials” improve syntax here.

Thanks for the observation, the suggested change was made.

Line 82. “to host organs with consequent repercussions” please define which kind of repercussion could be caused, especially patients outcomes.

We appreciate your observation, information related to the comment was included.

Line 86. please define clearly ʺautovaccineʺ

Information regarding the autovaccine concept was included.

Line 99 “previously isolated bacteria [12,13], which were selected considering the frequency with which they were isolated” please check the use of English, avoiding repetitions.

Moreover, I would suggest expanding the sentence concerning the goals of this study, clarifying side aspects.

English revision was carried out, and the requested information was also included.

Line 112. “strains” should this be in italics, as well?

We appreciate the observation; the correction was made.

Line 121. the sentence seems to end in this manner. Please, remove the space here after “between” and link with the following part of the sentence.

An inadvertent jump occurred; it was corrected.

 Line 126. “578.22 μg/mL” I suggest approximation to the first, max second decimal unit since we are talking about nanograms.

We appreciate the suggestion; the proposed changes were made.

Table 1 contains pH equal to 4 in all the compositions of lysate. This could be avoided by simply write it in the manuscript text presenting the table.

The proposal is very correct, the changes suggested in table 1 were made.

 Line 135. “To find out” I would substitute with “to define/determine.”

The substitution was made considering the suggestion.

 Line 211. “In the analysis” please check the syntax and the use of English.

The suggested analysis was performed, and the respective changes were made.

Concerning the discussion section, besides discussing the therapeutic approach for UTIs, consider mentioning other potential strategies, such as vaccine development, new drug delivery systems for vaccine delivery (liposomes, niosomes, foams) or alternative therapies like phage therapy. Discuss how finding alternative treatments for UTIs could improve patient outcomes and reduce healthcare costs.

Part of what was suggested about other alternatives for the control of UTI, was included in the discussion and expanded in the conclusions.

Line 320. “Studies that point out the” references are needed here.

The requested information was included, supported by specific references.

Line 363. “20 ?g/mL” the “?” should not be bold.

Thanks for the observation, the bold has been removed.

 Line 365. “Are due to the change of pH and it has been demonstrated that LPS treated with acid” too long sentence; please divide it.

The change was made.

 Conclusions need to be expanded, maybe moving some paragraphs or sentences from the very long discussion section. The definition of further perspectives could be also well appreciated.

The suggestion is very correct, some adjustments were made, we hope to comply with what was requested.

Comments on the Quality of English Language

Authors need to revise their paper: use of English is poor.

We sent the manuscript for style correction and language improvement.

Round 2

Reviewer 2 Report

Comments and Suggestions for Authors

Authors responded to my issues point by point. Now the paper can be published.